# Polyphenol Degradation Kinetics of Specialty Coffee in Different Presentations

**DOI:** 10.3390/foods14213600

**Published:** 2025-10-23

**Authors:** Frank Fernandez-Rosillo, Eliana Milagros Cabrejos-Barrios, Segundo Grimaldo Chávez-Quintana, Lenin Quiñones-Huatangari

**Affiliations:** 1Grupo de Modelamiento y Simulación de Procesos en la Industria Alimentaria, Instituto de Investigación de Ciencia de Datos, Universidad Nacional de Jaén, Cajamarca 06800, Peru; eliana_cabrejos@unj.edu.pe; 2Instituto de Investigación, Innovación y Desarrollo para el Sector Agrario y Agroindustrial (IIDAA), Facultad de Ingeniería y Ciencias Agrarias, Universidad Nacional Toribio Rodríguez de Mendoza, Amazonas 01001, Peru; segundo.quintana@untrm.edu.pe; 3Facultad de Ingeniería Zootecnista, Biotecnología, Agronegocios y Ciencia de Datos, Universidad Nacional Toribio Rodríguez de Mendoza, Amazonas 01001, Peru; lenin.quinones@untrm.edu.pe

**Keywords:** specialty coffee, polyphenols, reaction order, decay constant, activation energy, accelerated storage, Arrhenius, half-life

## Abstract

Polyphenols are chemical compounds found in plants, and coffee is an important source of them. The objective of the study was to evaluate the kinetics of polyphenol degradation in a blend of specialty coffee (green, roasted and roasted–ground beans), packaged in eight different packages, under accelerated storage conditions. The samples were stored at 40, 50 and 60 °C for 12, 8 and 4 days, respectively. The degradation kinetics were modelled based on chemical kinetics and determination of the reaction order. Using the Arrhenius model, the rate constants (*k*) and activation energies (*E_a_*) were estimated, which were then used to calculate and predict the half-life. The degradation followed zero-order kinetics. The rate constant (k) varied between 0.437 and 9.534 days^−1^ (40–60 °C). The *E_a_* ranged from 49.321 to 118.04 kJ*mol^−1^. The average shelf life shows a direct correlation with the characteristics and barrier properties of the packaging, with the longest storage times for daily storage at 25 °C being for vacuum-packed green beans (27.16 months), vacuum-packed roasted beans (3.14 months) and roasted ground coffee in trilaminate foil with a valve (40.21 months). Polyphenol retention decreased significantly with increasing temperature. For green bean, roasted bean and roasted ground coffee, the packaging that showed the best protection for the coffee was vacuum packaging and trilaminate with valve respectively, being crucial for preserving these bioactive compounds.

## 1. Introduction

Coffee is one of the world’s popular beverages, noted for its psychoactive effects, pleasant taste and an important source of polyphenols [1], compounds that possess several bioactive properties that can positively influence human health [2,3]. Specialty coffees are characterized by the uniqueness of their quality and origin, from the selection of the seedlings to the preparation of the beverage itself [4]. Flavor and aroma are the important properties for assessing coffee quality [5,6] and consumer preferences [7]. Coffee consists mainly of caffeine, diterpene, kahweol, chlorogenic acid and phenols. Among the phenolic compounds present, caffeic acid, cinnamic acid, chlorogenic acid, ferulic acid, coumaric acid, quercetol, kaempferol and isoquercitrin stand out [8,9], these compounds including volatiles, lipids, phenolic and alkaloids contribute to the basic taste sensation of coffee beverages, providing bitterness, astringency, strength and body to the coffee brew [10,11]. During roasting, simultaneous chemical reactions occur that promote the degradation of proteins, sugars, trigonelline, and chlorogenic acids [12], and the formation of substances resulting from Maillard reactions and Strecker degradation, which influence both the flavor and aroma of coffee [13,14]. Melanoidin is obtained from non-enzymatic roasting, which is responsible for the antioxidant activity of coffee [15]. The chlorogenic acid content decreases during this stage, while the melanoidin content increases [16].

Polyphenols are compounds with structures that oxidize easily, which can lead to the degradation of their antioxidant capacities [17,18]. There are two main forms of polyphenol degradation: enzymatic and non-enzymatic methods [19,20]. Enzymatic degradation involves enzymes such as polyphenol oxidase, lipoxygenase, and peroxidase [21]. Non-enzymatic degradation includes Maillard reactions, which cause phenol oxidation [22]. However, these are not the only ways in which polyphenols can be destroyed, but rather the main reactions that take place in food.

Studies show the presence of phenolic acids in green coffee (chlorogenic acid) and roasted coffee (other phenolic acids) [23,24], scientifically recognizing that coffee changes its chemical composition during the roasting process [14,25]. Roasted and ground coffee is considered a stable product [14]. The phenolic compounds in coffee are directly or indirectly related to the quality of the coffee in the cup [26], based on the presence of chlorogenic acids (6–12%) [27], which are responsible for the pigmentation, aroma formation, and astringency of coffee [28]. However, the thermal degradation of chlorogenic acids during roasting results in phenolic substances that contribute to bitterness [24].

It has been shown that the amount of polyphenols in coffee can vary significantly depending on the type of coffee, the roasting process, the extraction method, and storage conditions. In addition, factors such as temperature and packaging material play a crucial role in the preservation of these bioactive compounds [29]. Ref. [30] indicate that the rate of degradation of chlorogenic acids increases significantly with temperature during the storage of roasted coffee. Likewise, Refs. [31,32] show that the matrix, type of processing, and packaging are determining variables in the preservation of antioxidants. Ref. [12] demonstrated that the roasting process has a negative influence on bioactive compounds.

The quantification, analysis, and study of the stability of polyphenols in coffee are essential for understanding their antioxidant potential and health benefits, Therefore, this research aims to address a significant gap in current knowledge on the kinetics of polyphenol degradation in coffee, in particular by analyzing the effects of packaging and storage conditions on their stability. Although the variability of phenolic compounds in coffee has been widely documented, there is a lack of studies that systematically address how environmental factors influence the loss of these compounds over time. Furthermore, studying the kinetics of degradation in three states of coffee (green, roasted, and roasted ground) packaged in eight types of packaging commonly used in the coffee industry is crucial for the coffee industry in terms of quality.

Previous studies analyzing the stability of bioactive compounds present in coffee have reported that the degradation kinetics follow a zero-order reaction, indicating that the rate of loss does not depend on the concentration of the reactant [33,34,35], a debatable behavior that warrants evaluation to better understand the reaction mechanisms of phenolic compounds. The comparison of linear regressions and R^2^ between zero-order and first-order models plays a fundamental role in validating which model best describes the kinetics of polyphenol loss.

This study focuses on how different packaging and storage conditions affect the stability of polyphenols in green, roasted, and ground coffee beans, using the Folin–Ciocalteu method for measurement. This method is known for its ability to quantify phenolic compounds present in many matrices, including beverages and plant extracts, and is particularly appreciated for its practicality [36,37]. The objective of the study was to evaluate the kinetics of deterioration of total polyphenols in a blend of specialty coffee (*Coffea arabica*) in green beans, roasted beans, and ground roasted beans packaged in eight packages subjected to accelerated storage at temperatures of 40, 50, and 60 °C for 12, 8, and 4 days, respectively.

## 2. Materials and Methods

### 2.1. Materials

The sample consisted of 120 kg of a blend of specialty coffees (Blend Gourmet oro verde with 50% Geisha, 25% Caturra, and 25% Pache) with organic and quality certification, from the 2024 harvest of the Cooperativa Agraria Cafetalera Cenfrocafe in the province of Jaen, department of Cajamarca, Peru. Green coffee beans were purchased and divided into three parts: green beans, roasted beans, and ground roasted beans. The initial moisture content of the green beans was 10.5%. The samples were packaged for green beans in tocuyo bags (100% polycotton material with a light weight of 120 g/cm^2^) with low barrier properties, double-laminated aluminum foil (aluminum weight 92 g/m^2^) and vacuum-packed in Ecotac bags; for roasted coffee beans in vacuum-packed in Ecotac bags (High barrier laminated plastic structure with EVOH (Eth-ylene-Vinyl-Alcohol) layer and dimensions of 20 × 30 cm), and for roasted ground coffee in pressed cardboard boxes (made of mi-cro-corrugated, self-assembling brown kraft material, of dimensions 10 × 10 × 7 cm) with low barrier properties, bilaminate bag without valve with window and zipper (Kraft paper and aluminum of dimensions 14 × 22 cm, white, standard zipper closure, bilaminate bag (Kraft paper and aluminum) with degassing valve and zipper (dimen-sions 14 × 22 cm, Kraft color, standard zipper closure, weight 145 g/m^2^ and in a trilaminate bag (Kraft paper, aluminum, polyeth-ylene) with degassing valve and zipper (dimensions 25 × 12.3 cm). Each container held 250 g of sample. However, the cup profile of the coffee samples and the validation of their specialty status during accelerated storage were determined using the cupping method carried out by six Q Arabica Graders [38] (Figure 1).

### 2.2. Preparation of the Sample

A total of 120 kg of coffee was weighed using an electronic platform scale (T-Scale, TCS-K2, Taipei, Taiwan), which was distributed into three parts: for green coffee beans, 45 units per 250 g package were used, distributed as follows: 15 tocuyo packages, 15 bilaminated foil packages, and 15 vacuum-packed Ecotec bags, which were packaged using a vacuum packer (Grondoy, EVE, Jinhua, Zhejiang, China); to obtain roasted ground coffee, 15 kg of green coffee beans were placed in the roaster (Probat, P12-2, Memphis, TN, USA) at an initial temperature of 185 °C and a final temperature of 210 °C for 10.5 min under a medium roast. Once the process was complete, it was left to rest for 12 h. For packaging, 15 Ecotec bags of 250 g were used with a vacuum packer (Grondoy, EVE, China). For the roasted ground coffee, the roasting was carried out as described above and then ground using an industrial grinder (Ditting, KFA1403, Zürich, Switzerland). Fifteen kg of roasted coffee were placed in the hopper for 15 min with a grind setting of level 5, obtaining ground coffee. The roasted and ground coffee was then packaged in 60 containers (250 g each) distributed in 15 pressed cardboard boxes, 15 bilaminate containers without a valve, with a window and zipper, 15 bilaminate containers with a degassing valve and zipper, and 15 trilaminate coffee containers with a degassing valve and zipper. The packaged samples were sealed using a sealing machine (Ditting, FR-900, Zürich, Switzerland), with the exception of the pressed cardboard boxes and Ecotec vacuum bags. The experiment had three independent replicates for each combination of presentation, packaging, and coffee temperature.

### 2.3. Accelerated Storage

Once packaging was complete, each sample was coded and stored in three ovens (FAITHFUL, GX-45BE, Dongguan, China). Fifteen samples per container (120 containers) were placed in each oven at temperatures of 40, 50, and 60 °C over a period of 12 days for evaluation. Once the time for each temperature had elapsed, the samples were gradually removed from the ovens for roasting (in the case of green coffee bean samples) using a roaster (PROBAT-WERKE, BRZ, Emmerich am Rhein, Germany). 110 g of green coffee was placed at an initial temperature of 185 °C and a final temperature of 190 °C for a period of 15 min. Before grinding (for green coffee samples and roasted coffee samples), the coffee was left to rest for 10 min and then ground using a grinder (Mahlkönig, BRZ2, Hamburg, Germany). 11.55 g of coffee was placed in the hopper and ground for 3 s, with the grind size set to level 6.

### 2.4. Total Polyphenol Content

The total polyphenol content of each extract and its replicates was determined using the Folin–Ciocalteu method following the methodology of [39] with some modifications. Gallic acid was used as the standard: 10 mg was dissolved in distilled water to a total volume of 25 mL, obtaining the stock solution. From this, the calibration curve was prepared in the range of 0–140 mg/L (y = 0.0038x + 0.0051 with an R^2^ of 0.9781). For the extracts, 25 mg of ground coffee was weighed and mixed with 5 mL of ethanol/water (80:20 *v*/*v*). The mixture was stirred in an orbital shaker (Benchmark BT302, Suzhou, China) for 30 min at 3000 rpm and then centrifuged (Jiangsu Care Laboratory Instrument Co., Ltd., pro analytical CR 4000, Suzhou, Jiangsu, China) at 4000 rpm for 15 min. The supernatant was filtered using Whatman N° 42 paper. For analysis, 20 µL of the extract was pipetted into microplates, followed by 800 µL of distilled water, 40 µL of 10% Folin–Ciocalteu reagent, and 160 µL of 770 mM Na_2_CO_3_. Finally, the absorbance was measured at 765 nm using a Thermo Scientific™ Multiskan SkyHigh microplate reader with touchscreen. The total phenolic content is expressed as milligrams of gallic acid equivalents per litre of sample (mg GAE/L). All samples were analyzed in triplicate.

### 2.5. Polyphenol Degradation Model

#### 2.5.1. Chemical Kinetics

Equation (1) describes the main reaction of the polyphenol degradation model [40]:(1)$$\mathrm{Polyphenols}\;+\;\mathrm{Oxygen}\;\overset{k}{\to }\mathrm{ᴏ-Quinone}$$where *k* is the pseudo-constant reaction rate (h^−1^).

#### 2.5.2. Determination of the Reaction Order

The reaction order of chemical kinetics (zero or one order) was determined for the loss of desirable compounds (Equations (2)–(5)) to represent the above reaction, according to the result defined by the highest value of the coefficient of determination (R^2^) of the regression constructed with the data obtained from the decay of total polyphenols over time, for each storage temperature and type of container.(2)$$\mathrm{Order}\;0:\;\frac{d\left[C_{pt}\right]}{dt}=-k{\left[C_{pt}\right]}^{n}$$(3)$${\;\;\;\;\;\;\;\;\;\;\;\;\;\;\;\;\;\;\;\;\;C}_{pt}-C_{pt0}=-kt$$(4)$$\mathrm{Order}\;1:\;\frac{d\left[C_{pt}\right]}{dt}=-k{\left[C_{pt}\right]}^{n}$$(5)$$\;\;\;\;\;\;\;\;\;\;\;\;\;\;\;\;\;\;\;\;\left[C_{pt}\right]=\left[C_{pt0}\right]e^{-kt}$$
where *C_pt_* and *C_pt_*_0_ are the total polyphenol content (mg GAE/L) at a given time and at the start, respectively, *t* is time (days), *n* is the reaction order (zero or one) and *k* is the kinetic constant of deterioration in mg GAE/L/day or day^−1^ for order zero and one respectively.

Equations (3) and (5) were linearized, and graphs were plotted for [*C_pt_*] versus t (Equation (6)) and ln[*C_pt_*] versus *t* (Equation (7)). The rate constants *k* (1 or 2) were determined from the slopes of the linear graphs:(6)$$\mathrm{Order}\;0:\;C_{pt}=C_{pt0}-k_{1}t$$(7)$$\mathrm{Order}\;1:\;ln\left[C_{Pt}\right]=ln\left[C_{pt0}\right]-k_{2}t$$

#### 2.5.3. Calculation of Activation Energy (*E_a_*) and Pre-Exponential Factor (A_0_)

With the selected reaction order, the activation energy *E_a_* (kJ/mol) and the pre-exponential factor *A*_0_ (mg GAE/L/day) were estimated using an Arrhenius-type fit via Equation (10), constructed with the slopes of the straight line graphs between ln(*k*) and (1/(T + 273.15)) for each temperature and packaging in the study, where R is the gas constant (8.31446 J/mol°K) and T is the constant air temperature in storage (°C).(8)$$k=A_{0}e^{\left[-\left(\frac{E_{a}}{R}\right)\frac{1}{\left(T+273.15\right)}\right]}$$(9)$$ln\left[k\right]=\ln\left[A_{0}e^{\left[-\left(\frac{E_{a}}{R}\right)\frac{1}{\left(T+273.15\right)}\right]}\right]$$(10)$$ln\left[k\right]=\ln\left[A_{0\;}\right]-\left[\left(\frac{E_{a}}{R}\right)\frac{1}{\left(T+273.15\right)}\right]$$

#### 2.5.4. Estimation of the Average Retention Time of Total Polyphenols

Using Equation (11), the half-life (*t*_1/2_) (expressed in months) of total polyphenol retention in specialty coffee samples was calculated, following a zero-order reaction, where *C_pt_*_0_ is the initial polyphenol content in mg GAE/L, *C_pt_*_1/2_ represents 50% of the initial polyphenol content of each sample in mg GAE/L and *k*_1_ is the kinetic constant of deterioration in mg GAE/L/day.(11)$$\mathrm{Order}\;0:\;t_{1/2}=\frac{C_{pt0}-C_{pt1/2}}{k_{1}}$$

### 2.6. Statistical Analysis

The experiments were performed in triplicate, and the results were expressed using descriptive statistics with means ± standard deviations. In addition, data analysis was performed to determine whether there were significant differences in total polyphenol retention depending on the type of packaging, storage time, and temperature. For each storage temperature (40, 50, and 60 °C), a two-way analysis of variance (ANOVA) with replicates was applied, considering the type of packaging and storage time (days) as fixed factors. Subsequently, for multiple comparisons between the means of the different treatments (packages) at each measurement time, Tukey’s honestly significant difference (HSD) test was used, with a confidence level of 95% (α = 0.05). The results of this test are presented in the results tables using letters as superscripts, where means within the same sampling day that share a letter are not significantly different. The analyses were performed using the statistical software R (version 4.3.0, R Core Team, 2023) and the software used was Python (Version 3.11) through Google Colaboratory.

## 3. Results and Discussion

### 3.1. Validation of the Sample’s Specialty Status

Figure 1 shows the results of the sensory evaluation of the eight coffee packages, highlighting their ability to preserve quality during accelerated storage (40, 50, and 60 °C). The results showed that most samples maintained their specialty coffee status. Although natural sensory degradation was observed, it was not sufficient to cause the coffees in most packages to fall below the 80-point threshold. This phenomenon has been widely reported in previous studies, which show similar behaviour in the quality of speciality coffees under storage conditions [41,42,43].

The results of the sensory profile evaluation of coffee and the quantification of polyphenols show a correlation, which is an aspect of interest in the evaluation of coffee quality. Studies have shown that bioactive compounds such as polyphenols, especially chlorogenic acids, have a significant impact on the sensory characteristics of coffee [44,45] show that the concentration of volatile compound precursors is positively associated with a better sensory score, suggesting that a higher content is related to better cup quality. These findings support the idea that a higher presence of polyphenols and related compounds in coffee is associated with better sensory ratings, suggesting that polyphenols may be a relevant variable to consider in the development and evaluation of high-quality coffees [46].

### 3.2. Quantification of Total Polyphenols

Figure 2 shows the calibration curve for polyphenols with respect to absorbance, establishing the regression model y = 0.0038x + 0.0051 with an R^2^ of 0.9781.

Table 1, Table 2 and Table 3 show the results of polyphenol degradation during accelerated storage of coffee in three states and packaged in different types of packaging.

The results in Table 1 show that, as expected, there are no statistically significant differences in polyphenol content on day 0 between the different types of packaging at the start of the experiment (all share the letter ‘a’), but as time passes, the differences between packaging types become more evident. Green grain in BBFA packaging and vacuum packaging (VE1) generally showed greater polyphenol retention in the early days. Towards the end of storage (day 12), the trilaminate bag (BTCV) and vacuum-packed green beans (VE1) were the packaging types that retained significantly more polyphenols, unlike the cardboard box (CC) and tocuyo bag (BT), which showed some of the greatest degradation.

The results in Table 2 show that the trilaminate bag (BTCV) is the packaging that preserves significantly more polyphenols over time (days 4, 6, and 8), followed by the bilaminate bag with aluminum foil (BBFA) for green beans and the bilaminate bag with valve (BBCV) for roasted ground coffee. On the other hand, the tocuyo bag (BT) and the vacuum-packed roasted beans (VE2) showed the worst results at the end of the period (day 8), being significantly different from the best packaging.

The results in Table 3 show that, at this extreme temperature, the degradation of polyphenols is very rapid and severe. The differences between packaging types are marked. The bilaminate bag with valve (BBCV) for roasted ground coffee showed significant superiority in polyphenol retention at the end of storage (day 4), being the only one that remained above 89 mg GAE/L, while the bilaminated aluminum foil bag (BBFA) for green beans and the trilaminated bag (BTCV) showed some of the worst retention at this temperature, suggesting that their effectiveness may be compromised under extreme conditions.

Figure 3 shows descriptive statistics as a graphical summary of the decline in polyphenol content during accelerated storage for all coffee presentations, showing the temporal trends of polyphenols using a line graph for each temperature and evaluation day.

Figure 3 shows the temporal trend in polyphenol loss, indicating a downward curve where the slope indicates the rate of degradation. It should be noted that the steep slope in the early days suggests rapid initial loss, while a stabilizing trend indicates that most of the susceptible compounds have already been degraded. Storage temperature acts as a key accelerating factor, with curves at 60 °C showing a steeper slope than at 40 °C and a moderate slope at 50 °C. This behavior can be attributed to thermal stress accelerating the oxidation and decomposition of polyphenols. This trend was more pronounced in roasted and ground beans than in green beans. This phenomenon can be attributed to the thermal degradation of phenolic compounds during roasting and storage, thus affecting the antioxidant properties of coffee [29,37,47,48,49,50,51,52,53,54].

Furthermore, Figure 4 presents an overview of interactions using a heat map.

The heat map (Figure 4) shows that at 40 °C (yellow to blue scale), the darker blue-green tones at the beginning (day 0) correspond to higher concentrations of polyphenols (113–118 mg GAE/L). As storage progresses (day 12), lighter yellow-green areas appear, representing a decrease in total polyphenols, especially in BBFA (green coffee beans), VE2 (roasted coffee), CC, BBSV and BBCV (roasted–ground coffee) containers. BT, VE1 (vacuum-packed green coffee beans) and BTCV (trilaminate) maintained lighter colors for longer, indicating better retention and slower degradation. At 50 °C (yellow to brown scale) Orange and brown hues predominate at the beginning of storage, indicating high initial levels of polyphenols. Over time, they change to bright yellow tones, indicating intensified degradation. The BTCV and BBFA containers retained more yellow tones throughout, while BT and VE2 quickly turned light yellow, visually confirming the accelerated breakdown of polyphenols in these less protective materials. Moreover, at 60 °C (cold-hot scale: blue to red). Initially, blue tones indicate a high polyphenol content. As the days pass, the change to light blue and deep blue signifies a sharp decrease in polyphenols. The transition is particularly pronounced in BBFA, VE2, and BTCV, which turned blue early on, reflecting severe degradation on the fourth day. BBCV and CC showed slower transitions, maintaining cooler tones (light pink to blue), implying moderate stability. Consequently, containers with a resistant barrier (BTCV, VE1, BBCV) delay colour transition, indicating slower oxidation and better preservation of polyphenolic compounds, while permeable or unsealed materials (BT, CC) show rapid colour change, confirming greater exposure to oxygen and heat.

The Folin–Ciocalteu method used to quantify polyphenols in various coffee preparations has proven to be practical [46,55]. However, the method used (Folin–Ciocalteu) measures a total value that may include not only polyphenols but also other reducing compounds that could distort the results [56]. The interference of sugars and other non-phenolic compounds can lead to an overestimation of polyphenol levels when using this method, mainly due to its non-specific nature. This variability enables the use of more specialized methodologies such as high-performance liquid chromatography (HPLC) to obtain phenolic profiles, identifying and quantifying specific compounds and generating a more complete understanding of coffee’s antioxidant potential [50]. Comparative studies have shown that the values of phenolic compounds obtained by both methods can differ significantly. For example, research on coffee beans has revealed that the total phenol content, when measured using Folin–Ciocalteu, can be considerably higher than the values obtained using HPLC. This is partly due to the aforementioned interference, which highlights the need for a multidimensional approach to assessment [57,58].

Table 1 shows the observed losses in polyphenol content at 40 °C. It is noteworthy that green coffee beans maintained a relatively high polyphenol content compared to the other processed formats, which is consistent with the literature suggesting that polyphenols are more abundant in unprocessed or minimally processed coffee [29,59,60]. This phenomenon may be related to the fact that green grain has less exposure to environmental factors such as light and air, which can accelerate the degradation of phenolic compounds [61].

At higher storage temperatures, as shown in Table 3 (50 °C) and Table 4 (60 °C), the polyphenol content decreases progressively in all types of packaging, although the impact varies depending on the type of packaging. Bilinamized bags, for example, act as effective barriers against the loss of important compounds, resulting in greater polyphenol retention compared to less efficient packaging, such as paper [36,62].

It is also interesting to note that packaging with degassing valves appears to provide additional protection against oxidation, facilitating better preservation of polyphenols in roasted ground coffee [1,63]. The effectiveness of packaging could be associated with its oxygen permeability and moisture control, critical factors in the stability of phenolic content [62]. Vacuum packaging proved more efficient at retaining polyphenols in roasted coffee compared to other types of packaging such as aluminum foil bilaminate and trilaminate bags with degassing valves [64]. This superior performance of vacuum packaging for green and roasted beans is directly attributable to the almost total elimination of oxygen from the headspace, thus minimizing oxidative degradation. Similarly, the trilaminate foil bag with one-way valve for roasted ground coffee probably combines the barrier properties of the material with the valve’s function of releasing CO_2_ without allowing oxygen to enter, maintaining a low oxygen internal environment. The consistency of our kinetic results, showing the longest half-lives in these oxygen-restricted systems, strongly supports the validity of our model in capturing the dominant degradation pathway. This suggests that oxygen permeability and the ability to limit exposure to moisture are key factors in the stability of bioactive compounds in packaged coffee [9,63].

In conclusion, differences in polyphenol content when using different packaging suggest a detailed study of the barrier properties of each material, such as limiting exposure to oxygen and heat, which could provide additional information for preserving the polyphenol content of coffee. Therefore, the choice of packaging and storage conditions can have a significant impact on coffee quality, as well as on the health benefits associated with its consumption.

### 3.3. Determination of the Reaction Order

Table 4 shows the comparison of the regressions and R^2^ determined for the deterioration kinetics of polyphenols for zero and first order of the coffee samples under different storage conditions, which defined that the chemical reaction kinetics for the loss of phenolic compounds present in coffee exhibits a predominantly zero-order behavior.

The data obtained suggest that a zero-order model provides a better fit, indicating that the reaction of polyphenol loss under coffee storage conditions is constant over time, confirming that this phenomenon may be more related to physical factors such as temperature and exposure to oxygen than to the concentration of the polyphenols themselves [65,66]. These findings underscore the need for a consistent approach to coffee supply chain design, considering both bean quality and storage conditions to maximize the duration and effectiveness of its bioactive compounds and nutritional properties.

### 3.4. Determination of the Kinetic Constants of Deterioration (k)

Table 5 shows the Arrhenius equation determined for each coffee sample based on the values of the pre-exponential factor (*A*_0_) and calculated activation energy (*E_a_*), which can be used to predict the deterioration constants at different storage temperatures to which the different coffee presentations are subjected.

The total loss of polyphenols in coffee during storage is intrinsically related to several factors, including temperature, type of packaging, and the state of the bean (green, roasted, or ground). The Arrhenius model, which relates the reaction rate to temperature through *E_a_*, is fundamental to understanding these phenomena. In particular, its application in assessing the stability of polyphenols at different temperatures is crucial for determining the viability of storage and the long-term quality of coffee [63,67]. For example, it has been reported that packaging that limits access to O_2_, such as vacuum packaging, tends to show lower *E_a_* values, suggesting that reducing oxygen can inhibit the degradation of phenolic compounds, prolonging their effectiveness in terms of antioxidant properties [29,68]. Likewise, packaging that includes degassing valves (BBCV and BTCV) can have a significant impact on the preservation of polyphenols. Research has shown that the presence of carbon dioxide can protect polyphenols from degradation, suggesting that packaging methods that allow for the controlled release of gases can be beneficial [69]. Alternatively, packaging that allows gas exchange has shown greater loss of polyphenols at room temperature [70]. According to Arrhenius’ theory, at higher temperatures, the *E_a_* required for the loss of polyphenols tends to be lower, causing an acceleration in the degradation process.

The assumption of a constant activation energy (*E_a_*) in the Arrhenius model has limitations in complex food systems. In foods with multiple bioactive components, deterioration reactions do not always follow predictable linear patterns, leading to deviations in predictions. This simplification may obscure secondary reactions influenced by factors such as humidity, pH, or natural catalysts. For example, the degradation of phenolic compounds is accelerated by oxygen, an effect that is underestimated if *E_a_* is considered constant [71]. Furthermore, the model assumes equilibrium conditions that rarely occur in actual storage, where environmental fluctuations are frequent. Consequently, although the simplification offers useful predictions, shelf life estimates may be inaccurate [72]. Thus, the results should be interpreted with caution, considering the specific context of each food product.

Likewise, it was observed that optimized packaging, such as the Ecotac bag for green beans (*E_a_* of 107.67 kJ/mol*K), has higher *E_a_* values, suggesting greater resistance to polyphenol degradation compared to other packaging. This could be the result of the packaging’s ability to limit the exposure of polyphenols to oxygen and moisture, two critical factors in the oxidation process [72,73]. On the other hand, the trilaminate bag packaging with degassing valve and zipper (BTCV) for roasted ground coffee showed an *E_a_* value of 118.04 kJ/mol*K, which was the highest in the study. This result could indicate that the packaging design is particularly effective in retaining the quality of polyphenols, allowing for a gradual decrease in their oxidative degradation due to its sophisticated preservation methodology [74,75]. The high capacity of this packaging to minimize exposure to air and control O_2_ transfer is crucial for maintaining the stability of these compounds at higher temperatures [76]. In contrast, packaging with lower *E_a_* values, such as pressed cardboard boxes with 49,321 kJ/mol*K for roasted and ground coffee, appear to be more susceptible to rapid polyphenol degradation. This phenomenon can be attributed to its design, which may allow more direct contact with O_2_ and, therefore, an increase in the oxidation rate of polyphenols [77,78]. The permeability of cardboard can facilitate the diffusion of O_2_ and moisture into the packaging, accelerating the degradation process [79].

Furthermore, it can be observed that the variability of *E_a_* values between different forms of coffee (green beans, roasted beans, and ground coffee) is also clear. Green beans, which are more susceptible to repeated oxidation processes under high temperature conditions, may undergo slow degradation in more preferred packaging that adequately protects the chemical integrity of polyphenols [80,81]. In general, *E_a_* is a key parameter for understanding the stability of polyphenols in coffee in its different forms. A higher *E_a_* value implies greater resistance to polyphenol degradation, which means that more energy is required to initiate the degradation reaction. This indicates that polyphenols are less susceptible to environmental conditions that can induce their decomposition, such as high temperature, light, or the presence of oxygen. In the case of the coffee packaging analyzed, it has been observed that samples with a high *E_a_*, such as that obtained for roasted ground coffee in BTCV (118.04 kJ/mol), are more efficient at preserving polyphenols. This behavior is consistent with existing literature indicating that higher *E_a_* can reduce the degradation rate of bioactive compounds, which is essential for the nutritional and functional quality of coffee [82,83].

### 3.5. Estimated Half-Life of Total Polyphenol Retention

Table 6 shows the calculated values of the kinetic constants of deterioration (*k*) and half-lives (*t*_1/2_) (where half of the total polyphenol concentration is lost), following zero-order chemical kinetics, for coffee samples at the acceleration temperatures of the study (40, 50, and 60 °C) and at the everyday storage temperatures of supermarkets and homes (15, 20, and 25 °C).

The data in Table 6 show that green coffee in Ecotac bags (VE1) exhibits a t_1/2_ of 122.45 months at 15 °C, as opposed to 0.20 months at 60 °C. This phenomenon of accelerated degradation can be attributed to high temperatures, which potentially promote undesirable chemical reactions that lead to the loss of bioactive compounds such as polyphenols. The literature supports this observation, suggesting that proper temperature management during storage is essential for preserving phenolic compounds in coffee [29,51]. Bilaminate aluminum foil packaging (BBFA) and tocuyo bags (BT) demonstrate different levels of effectiveness in terms of *t*_1/2_. For example, green coffee beans in BBFA packaging have a shelf life of 10.63 months at 15 °C, which is significantly less than green coffee stored in vacuum packaging. These findings underscore how packaging material can affect the exposure of products to light and oxygen, factors that are known to contribute to the oxidation of polyphenols [3,63]. In the case of roasted coffee, it can be seen that the *t*_1/2_ values are considerably lower than those for green coffee. For example, coffee roasted in an Ecotac bag (VE2) has a *t*_1/2_ of only 6.82 months at 15 °C. The reduction in stability can be attributed to the roasting process, which on the one hand improves the sensory profile of coffee but, on the other, can lead to the degradation of phenolic compounds such as chlorogenic acids, which are vital for the antioxidant properties of coffee [56,68]. According to the evaluation of *t*_1/2_ at 60 °C, both roasted and green coffee beans exhibit extremely low polyphenol retention times. This indicates that, as the temperature increases, not only is faster oxidation promoted, but the formation of compounds is also favored which, although they may be more volatile and less desirable, can contribute to changes in the organoleptic characteristics of the beverage [28,59].

However, it is important to show the t_1/2_ values calculated at 25 °C (standard storage temperature) and the difference between packaging and presentations. in the case of packaging with green beans, BT, BBFA and VE1 exhibited t_1/2_ values of 7.3, 4.6 and 27.2 months, respectively, highlighting vacuum packaging (VE1) as the best packaging for the preservation of total polyphenols in this presentation. For the packaging of roasted beans VE2, the t_1/2_ was 3.2 months, which was less than that obtained in the three presentations with green beans. In the group of packages containing roasted ground coffee CC, BBSV, BBCV, and BTCV, the calculated t_1/2_ values were 4.5, 4.4, 5.5, and 40.2 months, respectively, suggesting that BTCV offers the best conditions for preserving the bioactive compounds in coffee over time, not only for ground roasted coffee but also in comparison with the other types of packaging.

The *k* analysis provides more context for this data. This parameter represents the amount of total polyphenols in mg GAE/L that is lost in the product (coffee) in one day of storage. It can be observed that *k* increases considerably with increasing temperature for all packaging, for example, for VE1 and BTCV (packing’s with higher *t*_1/2_), increases from 0.0149565 to 6.4372803 and 0.0085808 to 6.6204522 mg GAE/L/day from 15 to 60 °C, respectively, and in the case of BBSV and VE2 (packing’s with lower *t*_1/2_), values of 0.1668885 to 7.2057031 and 0.2592307 to 5.8794484, respectively, were observed, showing differences for *k* between packing’s with higher and lower *t*_1/2_ at low temperatures and confirming a direct relationship between temperature and the rate of degradation of polyphenols. From the analysis presented, it can be inferred that low k values will result in prolonged shelf life. This pattern is consistent with various studies that highlight that increased temperature accelerates the kinetics of chemical oxidation reactions, which are responsible for the decline in coffee quality [28,56,68].

The results of this study should be interpreted in light of certain limitations. First, they include the Folin–Ciocalteu method used, which does not discriminate between different types of phenols, preventing the identification and quantification of specific polyphenol compounds present in coffee. This limits the ability to perform a detailed analysis of which compounds are most susceptible to degradation and which retain their stability depending on packaging and storage conditions. In addition, the method presents possible interference due to the presence of substances such as sugars and proteins that can interfere with the reaction and affect the accuracy of the results. However, as discussed in the research, this problem is considered marginal because the study consisted of determining the total polyphenols in the samples and not identifying and quantifying specific polyphenol compounds. Secondly, in the packaging evaluated, the lack of data on oxygen and water vapor transfer rates is a limitation, as these parameters influence the interaction of the packaging contents with the external environment and could affect the stability of the polyphenols. However, this limitation leads to a possible bias in the evaluation of polyphenol retention in packaging. Finally, the use of accelerated tests to assess the stability of polyphenols in coffee may not accurately reflect actual storage conditions at lower temperatures. The controlled temperature and humidity conditions in these tests may not correctly simulate the behavior of polyphenols in conventional storage environments, which may lead to an overestimation of the stability of bioactive compounds under more natural conditions.

## 4. Conclusions

The evaluation of the kinetics of polyphenol deterioration in different forms of specialty coffee and its correlation with various packaging systems has provided fundamental information on the stability and preservation of these bioactive compounds under accelerated storage conditions. The polyphenol loss reaction fits a zero-order kinetic model, indicating that the decay of bioactive compounds under specialty coffee storage conditions is constant over time, suggesting that this phenomenon may be more related to external physical factors such as temperature and oxygen exposure than to the concentration of the polyphenols themselves. The degradation rate constants found reflect an inverse relationship between temperature and shelf life. The data indicate that packaging such as aluminum trilaminate bags and vacuum bags proved effective in controlled environments, while cardboard containers showed lower polyphenol retention. Therefore, this study highlights the complexity of the degradation kinetics of polyphenols in coffee and the critical influence of storage and packaging systems. These findings are highly relevant to the coffee industry, suggesting that the appropriate selection of packaging and optimal storage conditions can significantly extend the shelf life of polyphenols, thus contributing to the preservation of coffee quality in the market. It is also crucial to consider various lines of research that could provide a deeper understanding of the phenomenon. One suggestion would be to investigate the degradation of specific polyphenol compounds through detailed studies using advanced chemical analysis techniques. Exploring innovations in packaging that combine different materials or incorporate bioactive additives could further optimize the preservation of these critical compounds. Alongside these studies on packaging efficiency, it would be valuable to conduct sensory evaluations that correlate polyphenol retention with consumer perception. Consequently, these future initiatives would not only contribute to the field of food science and process engineering, but could also influence commercial and marketing decisions in the coffee industry.

## Figures and Tables

**Figure 1 foods-14-03600-f001:**
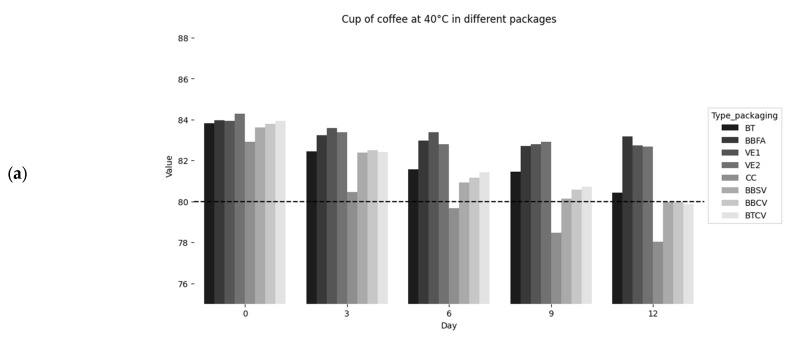
Cup scores for each presentation and packaging during accelerated storage at 40 °C (**a**), 50 °C (**b**), and 60 °C (**c**). BT: tocuyo bag. BBFA: bilaminated aluminum foil bag. VE1: Ecotac vacuum bag with green beans. VE2: Ecotac vacuum bag with roasted beans. CC: pressed cardboard box. BBSV: bilaminated bag without valve, with window and zipper. BBCV: bilaminated bag with degassing valve and zipper. BTCV: trilaminate bag with degassing valve and zipper.

**Figure 2 foods-14-03600-f002:**
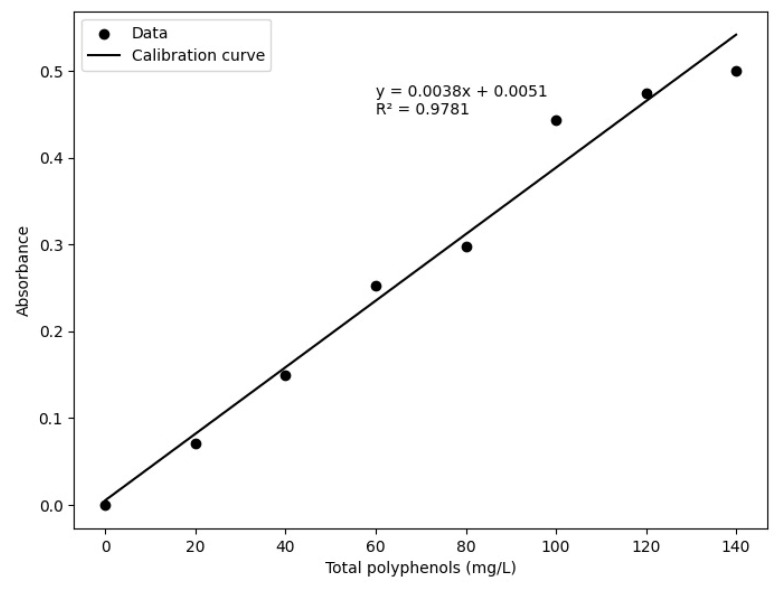
Calibration curve for total polyphenols.

**Figure 3 foods-14-03600-f003:**
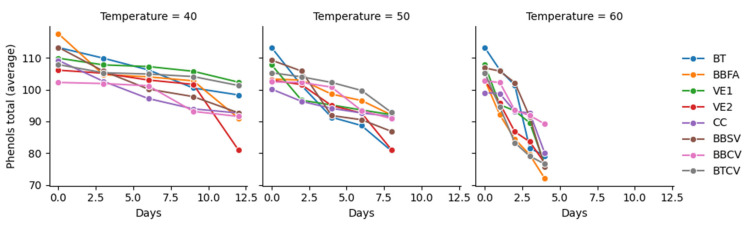
Temporal trends in total polyphenols in specialty coffee samples under accelerated storage conditions. BT: Tocuyo bag with green beans. BBFA: Bilaminated aluminum bag with green beans. VE1: Ecotac vacuum bag with green beans. VE2: Ecotac vacuum bag with roasted beans. CC: Pressed cardboard box with ground roasted coffee. BBSV: Bilaminated bag without valve, with window and zipper, with ground roasted coffee. BBCV: bilaminate bag with degassing valve and zipper with ground roasted coffee. BTCV: trilaminate bag with degassing valve and zipper with ground roasted coffee.

**Figure 4 foods-14-03600-f004:**
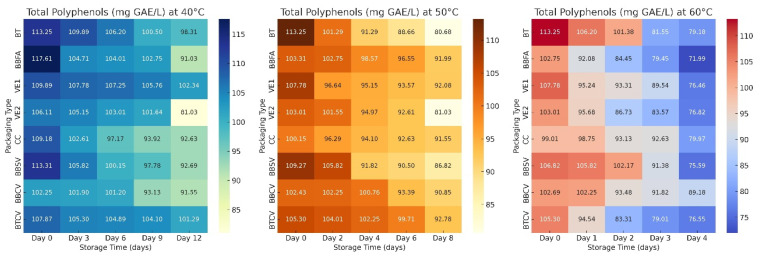
Heat map of total polyphenol degradation in specialty coffee during accelerated storage. BT: Tocuyo bag with green beans. BBFA: Bilaminated aluminum bag with green beans. VE1: Ecotac vacuum bag with green beans. VE2: Ecotac vacuum bag with roasted beans. CC: Pressed cardboard box with ground roasted coffee. BBSV: Bilaminated bag without valve, with window and zipper, with ground roasted coffee. BBCV: bilaminate bag with degassing valve and zipper with ground roasted coffee. BTCV: trilaminate bag with degassing valve and zipper with ground roasted coffee.

**Table 1 foods-14-03600-t001:** Total polyphenol values during storage at 40 °C.

Days	Total Polyphenol Content (mg GAE/L)							
	T 40 °C/HR 65%							
	Green Bean			Roasted Bean	Ground Roast			
	BT	BBFA	VE1	VE2	CC	BBSV	BBCV	BTCV
0	113.25 ± 0.02 ^a^	117.61 ± 0.01 ^a^	109.89 ± 0.01 ^a^	106.11 ± 0.00 ^a^	109.18 ± 0.02 ^a^	113.31 ± 0.09 ^a^	102.25 ± 0.02 ^a^	107.87 ± 0.01 ^a^
3	109.89 ± 0.02 ^a^	104.71 ± 0.00 ^b^	107.78 ± 0.03 ^a^	105.15 ± 0.01 ^ab^	102.61 ± 0.01 ^c^	105.82 ± 0.02 ^b^	101.90 ± 0.00 ^c^	105.30 ± 0.03 ^ab^
6	106.20 ± 0.01 ^a^	104.01 ± 0.01 ^a^	107.25 ± 0.02 ^a^	103.01 ± 0.03 ^ab^	97.17 ± 0.02 ^d^	100.15 ± 0.01 ^c^	101.20 ± 0.05 ^bc^	104.89 ± 0.02 ^a^
9	100.50 ± 0.01 ^b^	102.75 ± 0.02 ^a^	105.76 ± 0.01 ^a^	101.64 ± 0.01 ^b^	93.92 ± 0.03 ^d^	97.78 ± 0.01 ^c^	93.13 ± 0.02 ^d^	104.10 ± 0.01 ^a^
12	98.31 ± 0.01 ^b^	91.03 ± 0.02 ^c^	102.34 ± 0.02 ^a^	81.03 ± 0.02 ^e^	92.63 ± 0.02 ^c^	92.69 ± 0.02 ^c^	91.55 ± 0.01 ^c^	101.29 ± 0.01 ^a^

BT: tocuyo bag. BBFA: bilaminated aluminum foil bag. VE1: Ecotac vacuum bag with green beans. VE2: Ecotac vacuum bag with roasted beans. CC: pressed cardboard box. BBSV: bilaminated bag without valve, with window and zipper. BBCV: bilaminated bag with degassing valve and zipper. BTCV: trilaminate bag with degassing valve and zipper. ±: Mean Standard Deviation for 6 repetitions. Different letters represent significant differences between treatments according to Tukey’s test (*p* < 0.05).

**Table 2 foods-14-03600-t002:** Total polyphenol values during storage at 50 °C.

Days	Total Polyphenol Content (mg GAE/L)							
	T 50 °C/HR 65%							
	Green Bean			Roasted Bean	Ground Roast			
	BT	BBFA	VE1	VE2	CC	BBSV	BBCV	BTCV
0	113.25 ± 0.02 ^a^	103.31 ± 0.01 ^c^	107.78 ± 0.03 ^b^	103.01 ± 0.03 ^c^	100.15 ± 0.01 ^d^	109.27 ± 0.01 ^b^	102.43 ± 0.04 ^c^	105.30 ± 0.03 ^c^
2	101.29 ± 0.01 ^c^	102.75 ± 0.02 ^bc^	96.64 ± 0.00 ^e^	101.55 ± 0.00 ^c^	96.29 ± 0.02 ^e^	105.82 ± 0.02 ^a^	102.25 ± 0.02 ^bc^	104.01 ± 0.03 ^ab^
4	91.29 ± 0.00 ^e^	98.57 ± 0.02 ^c^	95.15 ± 0.01 ^d^	94.97 ± 0.01 ^d^	94.10 ± 0.01 ^d^	91.82 ± 0.02 ^e^	100.76 ± 0.02 ^b^	102.25 ± 0.03 ^a^
6	88.66 ± 0.03 ^d^	96.55 ± 0.02 ^b^	93.57 ± 0.01 ^c^	92.61 ± 0.01 ^c^	92.63 ± 0.02 ^c^	90.50 ± 0.03 ^d^	93.39 ± 0.00 ^c^	99.71 ± 0.01 ^a^
8	80.68 ± 0.02 ^e^	91.99 ± 0.04 ^b^	92.08 ± 0.01 ^b^	81.03 ± 0.01 ^e^	91.55 ± 0.01 ^b^	86.82 ± 0.03 ^d^	90.85 ± 0.03 ^c^	92.78 ± 0.03 ^b^

BT: tocuyo bag. BBFA: bilaminated aluminum foil bag. VE1: Ecotac vacuum bag with green beans. VE2: Ecotac vacuum bag with roasted beans. CC: pressed cardboard box. BBSV: bilaminated bag without valve, with window and zipper. BBCV: bilaminated bag with degassing valve and zipper. BTCV: trilaminate bag with degassing valve and zipper. ±: Mean Standard Deviation for 6 repetitions. Different letters represent significant differences between treatments according to Tukey’s test (*p* < 0.05).

**Table 3 foods-14-03600-t003:** Total polyphenol values during storage at 60 °C.

Days	Total Polyphenol Content (mg GAE/L)							
	T 60 °C/HR 65%							
	Green Bean			Roasted Bean	Ground Roast			
	BT	BBFA	VE1	VE2	CC	BBSV	BBCV	BTCV
0	113.25 ± 0.02 ^a^	102.75 ± 0.02 ^c^	107.78 ± 0.03 ^b^	103.01 ± 0.03 ^c^	99.01 ± 0.03 ^d^	106.82 ± 0.03 ^b^	102.69 ± 0.03 ^c^	105.30 ± 0.03 ^bc^
1	106.20 ± 0.01 ^a^	92.08 ± 0.04 ^e^	95.24 ± 0.03 ^d^	95.68 ± 0.01 ^d^	98.75 ± 0.02 ^c^	105.82 ± 0.02 ^a^	102.25 ± 0.02 ^b^	94.54 ± 0.01 ^de^
2	101.38 ± 0.02 ^a^	84.45 ± 0.00 ^f^	93.31 ± 0.02 ^c^	86.73 ± 0.03 ^ef^	93.13 ± 0.01 ^cd^	102.17 ± 0.04 ^a^	93.48 ± 0.02 ^c^	83.31 ± 0.01 ^f^
3	81.55 ± 0.01 ^d^	79.45 ± 0.03 ^d^	89.54 ± 0.01 ^b^	83.57 ± 0.05 c^d^	92.63 ± 0.02 ^a^	91.38 ± 0.01 ^a^	91.82 ± 0.06 ^a^	79.01 ± 0.02 ^d^
4	79.18 ± 0.01 ^c^	71.99 ± 0.00 ^d^	76.46 ± 0.01 ^c^	76.82 ± 0.00 ^c^	79.97 ± 0.00 ^c^	75.59 ± 0.00 ^c^	89.18 ± 0.04 ^a^	76.55 ± 0.01 ^c^

BT: tocuyo bag. BBFA: bilaminated aluminum foil bag. VE1: Ecotac vacuum bag with green beans. VE2: Ecotac vacuum bag with roasted beans. CC: pressed cardboard box. BBSV: bilaminated bag without valve, with window and zipper. BBCV: bilaminated bag with degassing valve and zipper. BTCV: trilaminate bag with degassing valve and zipper. ±: Mean Standard Deviation for 6 repetitions. Different letters represent significant differences between treatments according to Tukey’s test (*p* < 0.05).

**Table 4 foods-14-03600-t004:** Determination of the reaction order.

Sample	Packaging	Order	Temperature			
			40 °C	50 °C	60 °C	Selection
			Regression (R^2^)	Regression (R^2^)	Regression (R^2^)	
Green bean	BT	0	y = −1.309x + 113.48 (0.9858)	y = −3.8889x + 110.59 (0.9576)	y = −9.2778x + 114.87 (0.9341)	0
		1	y = −0.0124x + 4.733 (0.9849)	y = −0.0406x + 4.7097 (0.9706)	y = −0.098x + 4.7533 (0.9237)	
	BBFA	0	y = −1.837x + 115.04 (0.8546)	y = −1.4415x + 104.4 (0.9526)	y = −7.4152x + 100.97 (0.9835)	1
		1	y = −0.018x + 4.7475 (0.8549)	y = −0.0147x + 4.6494 (0.9486)	y = −0.0859x + 4.6204 (0.9920)	
	VE1	0	y = −0.570x + 110.03 (0.9303)	y = −1.7237x + 103.94 (0.763)	y = −6.8333x + 106.13 (0.9200)	0
		1	y = −0.005x + 4.7011 (0.9261)	y = −0.0174x + 4.643 (0.7781)	y = −0.0748x + 4.6704 (0.9119)	
Roasted bean	VE2	0	y = −1.7895x + 110.13 (0.6643)	y = −2.6462x + 105.22 (0.9109)	y = −6.4503x + 102.06 (0.9814)	0
		1	y = −0.0191x + 4.7089 (0.6466)	y = −0.0286x + 4.6609 (0.8929)	y = −0.0722x + 4.6296 (0.9867)	
Ground roast	CC	0	y = −1.3928x + 107.46 (0.9369)	y = −1.0424x + 99.114 (0.9356)	y = −4.4181x + 101.54 (0.8184)	1
		1	y = −0.0139x + 4.6777 (0.9457)	y = −0.0109x + 4.5964 (0.9417)	y = −0.0491x + 4.6246 (0.8003)	
	BBSV	0	y = −1.6423x + 111.8 (0.9698)	y = −3.0117x + 108.89 (0.9044)	y = −7.6901x + 111.73 (0.8585)	1
		1	y = −0.016x + 4.7182 (0.9773)	y = −0.0308x + 4.6922 (0.9108)	y = −0.0838x + 4.7276 (0.8343)	
	BBCV	0	y = −1.0058x + 104.04 (0.8365)	y = −1.6009x + 104.34 (0.8713)	y = −3.7456x + 103.38 (0.9101)	0
		1	y = −0.0104x + 4.6461 (0.8353)	y = −0.0165x + 4.6492 (0.8692)	y = −0.039x + 4.6394 (0.9151)	
	BTCV	0	y = −0.4786x + 107.56 (0.9192)	y = −1.4664x + 106.68 (0.8766)	y = −7.3012x + 102.34 (0.9264)	0
		1	y = −0.0046x + 4.6782 (0.9181)	y = −0.0148x + 4.6713 (0.8651)	y = −0.0817x + 4.6306 (0.9421)	

BT: tocuyo bag. BBFA: bilaminated aluminum foil bag. VE1: Ecotac vacuum bag with green beans. VE2: Ecotac vacuum bag with roasted beans. CC: pressed cardboard box. BBSV: bilaminated bag without valve, with window and zipper. BBCV: bilaminated bag with degassing valve and zipper. BTCV: trilaminate bag with degassing valve and zipper.

**Table 5 foods-14-03600-t005:** *E_a_*, *A*_0_ and *k* values for the degradation of total polyphenols in coffee for each sample and package.

Sample	Packaging	*E_a_* (kJ/mol*K)	*A*_0_ (mg GAE/L/day)	Arrhenius Equation Determined
Green bean	BT	85.085	2.06 × 10^14^	$k=2.06E+14e^{\left[-\left(\frac{85.085}{R(T+273.15)}\right)\right]}$
	BBFA	59.714	1.22 × 10^10^	$k=1.22E+10e^{\left[-\left(\frac{59.714}{R(T+273.15)}\right)\right]}$
	VE1	107.67	4.82 × 10^17^	$k=4.82E+17e^{\left[-\left(\frac{107.67}{R(T+273.15)}\right)\right]}$
Roasted bean	VE2	55.419	2.85 × 10^9^	$k=2.85E+09e^{\left[-\left(\frac{55.419}{R(T+273.15)}\right)\right]}$
Ground roast	CC	49.321	1.75 × 10^8^	$k=1.75E+08e^{\left[-\left(\frac{49.321}{R(T+273.15)}\right)\right]}$
	BBSV	66.850	2.16 × 10^11^	$k=2.16E+11e^{\left[-\left(\frac{66.850}{R(T+273.15)}\right)\right]}$
	BBCV	56.886	2.86 × 10^9^	$k=2.86E+09e^{\left[-\left(\frac{56.886}{R(T+273.15)}\right)\right]}$
	BTCV	118.04	2.09 × 10^19^	$k=2.09E+19e^{\left[-\left(\frac{118.04}{R(T+273.15)}\right)\right]}$

BT: tocuyo bag. BBFA: bilaminated aluminum foil bag. VE1: Ecotac vacuum bag with green beans. VE2: Ecotac vacuum bag with roasted beans. CC: pressed cardboard box. BBSV: bilaminated bag without valve, with window and zipper. BBCV: bilaminated bag with degassing valve and zipper. BTCV: trilaminate bag with degassing valve and zipper.

**Table 6 foods-14-03600-t006:** Half-life times (*t*_1/2_) for the degradation of total polyphenols.

Sample	Packaging	Storage Temperature (°C)	*k* (mg GAE/L/Day)	*t*_1/2_ Order Zero (Months)
Green bean	BT	15	0.0790631	23.9
		20	0.1448005	13.0
		25	0.2598701	7.3
		40	1.3429945	1.4
		50	3.6884747	0.5
		60	9.5343346	0.2
	BBFA	15	0.1844677	10.6
		20	0.2820690	7.0
		25	0.4252168	4.6
		40	1.3465856	1.5
		50	2.7363711	0.7
		60	5.3289017	0.4
	VE1	15	0.0149565	122.5
		20	0.0321659	56.9
		25	0.0674234	27.2
		40	0.5388931	3.4
		50	1.9353609	1.0
		60	6.4372803	0.3
Roasted bean	VE2	15	0.2592307	6.8
		20	0.3844649	4.6
		25	0.5627154	3.2
		40	1.6402242	1.1
		50	3.1673386	0.6
		60	5.8794484	0.3
Ground roast	CC	15	0.2026485	9.0
		20	0.2877919	6.3
		25	0.4039308	4.5
		40	1.0466383	1.7
		50	1.8799240	1.0
		60	3.2600315	0.6
	BBSV	15	0.1668885	11.3
		20	0.2684727	7.1
		25	0.4250620	4.4
		40	1.5448715	1.2
		50	3.4168621	0.6
		60	7.2057031	0.3
	BBCV	15	0.1413287	12.1
		20	0.2118019	8.1
		25	0.3131404	5.5
		40	0.9389597	1.8
		50	1.8450157	0.9
		60	3.4813704	0.5
	BTCV	15	0.0085808	209.5
		20	0.0198655	90.5
		25	0.0447145	40.2
		40	0.4365327	4.1
		50	1.7730255	1.0
		60	6.6204522	0.3

BT: tocuyo bag. BBFA: bilaminated aluminum foil bag. VE1: Ecotac vacuum bag with green beans. VE2: Ecotac vacuum bag with roasted beans. CC: pressed cardboard box. BBSV: bilaminated bag without valve, with window and zipper. BBCV: bilaminated bag with degassing valve and zipper. BTCV: trilaminate bag with degassing valve and zipper.

## Data Availability

The original contributions presented in this study are included in this article. For further information, please contact the corresponding author.

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
