# Peer review of "Polyphenol Degradation Kinetics of Specialty Coffee in Different Presentations"

_foods, 2025, doi:10.3390/foods14213600_

Round 1

Reviewer 1 Report

Comments and Suggestions for Authors

This paper presents an interesting and important study on the degradation of polyphenols in specialty coffee. The research is well designed, and the findings have practical implications for the coffee industry. However, there are several areas where the manuscript could be strengthened to improve as follow.

  • The abstract effectively summarizes the study's objective, methodology, and key findings. However, it should be more concise and impactful. The sentence "The degradation kinetics was modelled by determining the reaction order, rate constants (k), activation energies (En) using the Arrhenius model, and half-life" is a bit clunky. It could be rephrased to flow better. Also, the half-life data is presented for three specific cases. It would be helpful to include a more general statement about the trends observed before listing the specific values.
  • The introduction effectively establishes the importance of coffee as a source of polyphenols and its health benefits. It highlights the significance of specialty coffee and the factors influencing its quality, such as flavor and aroma.
  • The manuscript could better connect the general information about coffee polyphenols to the specific research question. For example, the statement, "The composition of coffee consists mainly of caffeine, diterpene, kahweol, chlorogenic acid..." is broad. It should be followed by a sentence that directly links these compounds to the study.
  • The discussion on polyphenol degradation mechanisms (enzymatic and non-enzymatic) provides a good foundation for understanding the study's context.
  • The literature review adequately supports the claim that polyphenol content varies with coffee type, roasting, extraction, and storage conditions, setting the stage for the current research.
  • The clear statement of the research objective—to evaluate the kinetics of total polyphenol deterioration in different coffee presentations and packaging under accelerated storage—is commendable.
  • While the introduction mentions the Folin-Ciocalteu method, a brief discussion on its limitations or potential interferences (e.g., other reducing substances reacting with the reagent) would add nuance and demonstrate a deeper understanding of the analytical technique. This is particularly relevant given the complexity of coffee matrices.
  • The transition from general information about coffee and polyphenols to the specific focus on degradation kinetics could be smoother. Perhaps a paragraph explicitly stating the gap in current knowledge that this study aims to fill, beyond just the general variability of polyphenols, would be beneficial.
  • While it's mentioned that 15 samples per container (120 containers) were used and experiments were performed in triplicate, the exact experimental design could be clarified. For instance, how many independent replicates were there for each combination of coffee presentation, packaging, and temperature? Clarifying this would enhance the statistical robustness.
  • For the roasted and roasted-ground coffee, more details on the roasting profile (e.g., roast degree, color measurements like Agtron or L*a*b* values) would be beneficial. This is important as roasting significantly impacts polyphenol content and composition.
  • While the method is standard, mentioning any internal validation steps (e.g., linearity range, detection limits, precision, accuracy) performed for this specific coffee matrix would strengthen the analytical rigor.
  • In equations (2) and (4), Cpt and Cpt0 are defined as total polyphenol content (mg/L). However, the Folin-Ciocalteu method typically measures total phenolic content, often expressed as gallic acid equivalents (GAE). Clarifying if the results are indeed in mg/L of actual polyphenols or GAE would be important for interpretation. If GAE, it should be explicitly stated.
  • Briefly discussing the assumptions of the Arrhenius model in the context of food degradation kinetics would be valuable. For example, assuming a constant activation energy over the temperature range studied.
  • Figure 1 is mentioned in the methods section but the data itself is not presented in the results section. While cup scores are not directly related to polyphenol degradation kinetics, their inclusion in the methods implies some relevance. If it's not discussed in the results, it might be better to either remove its mention from the methods or briefly explain its role in validating specialty status and how it correlates (or doesn't) with polyphenol retention.
  • While Figure 3 (temporal trends) and Figure 4 (heat map) are mentioned, they are not included in the provided text. Assuming they are visual representations of the data in Tables 1-3, ensuring they are clear, properly labeled, and easy to interpret is crucial. The heat map should clearly distinguish between different coffee presentations and packaging types.
  • The Ea and k values are presented; a more in-depth discussion of their practical implications would be beneficial. For example, comparing Ea values across different packaging types and coffee presentations to explain *why* certain combinations offer better protection. What do the specific ranges of k values (0.437 to 9.534 days⁻¹) mean in terms of shelf-life for consumers?
  • The abstract mentions predicted longest half-lives, but the detailed discussion in the results section could elaborate more on these predictions. For instance, a table summarizing the half-life values for all conditions at a standard storage temperature (e.g., 25 °C) would be very impactful. This would directly address the practical utility of the kinetic models.
  • While the discussion touches upon oxygen permeability and moisture control, a deeper dive into the specific mechanisms by which different packaging materials protect polyphenols would be valuable. For example, how do the layers in bilaminate/trilaminate bags contribute to barrier properties? What specific compounds are likely degrading (e.g., chlorogenic acids)?
  • A brief acknowledgment of the study's limitations would enhance its credibility. For example, the use of accelerated storage, while necessary for kinetic studies, might not perfectly mimic long-term ambient storage conditions. Also, the Folin-Ciocalteu method measures total phenolics, not individual polyphenol compounds, which could have different degradation kinetics.
  • The conclusion could be slightly expanded to include a forward-looking statement or suggestions for future research. For example, investigating the degradation of specific polyphenol compounds, exploring other packaging innovations, or conducting sensory evaluations alongside chemical analysis to correlate polyphenol retention with consumer perception.

Reviewer 2 Report

Comments and Suggestions for Authors

Reviewer Report

Manuscript title: Polyphenol Degradation Kinetics of Specialty Coffee in Different Presentations

General Assessment

This manuscript addresses an interesting topic in food chemistry, namely the stability of polyphenols in specialty coffee under different packaging and accelerated storage conditions. The research question is of high relevance for both scientific and industrial audiences. The authors provide many experimental data on total polyphenol content, kinetic modeling, and packaging effects.

However, despite the potential contribution, there are several weaknesses that require attention before the manuscript can be considered for publication. My detailed evaluation is provided below.

  1. Research question
    • The objective of the study is described in the Abstract (lines 18–29) and reiterated in the Conclusions (lines 414–424). While clear, it is not explicitly framed as a hypothesis or research gap. The Introduction (lines 57–91) remains largely descriptive and should clearly articulate what specific gap in the literature this study fills and why the combination of three coffee states and eight packaging types is scientifically novel.
  2. Experimental setup
    • The Materials and Methods section is detailed (lines 94–137 for packaging, lines 149–160 for polyphenol assay, and lines 162–185 for kinetic modeling). Nevertheless, there are shortcomings:
      • Only one packaging type (Ecotac bags) has reported oxygen transmission rate (OTR) values (line 101–103). For a comparative study of packaging performance, permeability parameters should be provided for all tested materials.
      • The use of triplicates (line 186) may be statistically limited for kinetic modeling. A justification for this choice should be included.
  1. Operating parameters
    • Storage was conducted at 40, 50, and 60 °C under 65% relative humidity (lines 133–137). While appropriate for accelerated testing, the oxygen concentration within packages was not monitored. Given that polyphenol degradation is highly oxygen-dependent, this omission weakens the reliability of the kinetic analysis.
    • Polyphenol quantification relies solely on the Folin–Ciocalteu assay (lines 149–160). This method is non-specific and can overestimate polyphenols due to interference from other reducing compounds. Validation by HPLC or at least a cross-reference to literature values for chlorogenic acid is strongly recommended.
  2. Logical inconsistencies
    • Line 46–47: “Mallard reactions” should be corrected to Maillard reactions.
    • Lines 74–91: The Introduction states that polyphenol loss is concentration-independent and thus zero-order, yet the Methods (lines 166–175) describe testing both zero- and first-order models. The text should be harmonized to avoid contradiction.
    • Lines 213–221: The authors claim “no significant differences on day 0,” while Table 1 shows initial values ranging from 109.89 to 117.61 mg/L. This should be rephrased as “no statistically significant differences” to avoid misleading the reader.
    • Lines 333–349: The interpretation of higher activation energy (Ea) values as implying greater resistance to degradation is not consistently applied throughout the discussion. Clarification is required to maintain logical coherence.
  3. Points requiring substantial improvement
    • Provide full packaging characteristics (OTR, WVTR) for all materials tested.
    • Include statistical dispersion (standard deviations or confidence intervals) for kinetic constants in Tables 5 and 6 (lines 327–380). Currently, only mean values are shown, which undermines reproducibility.
    • Figures 3 and 4 (lines 255–276) lack error bars, which are essential to judge variability.
    • Strengthen the Introduction (lines 57–91) with a concise statement of the research gap compared to recent literature.
    • Ensure consistency of terminology: e.g., replace vague phrases such as “polyphenol retention times” (line 399–401) with the precise “half-life (t₁/₂).”

Minor Comments

  • Some sentences in the Discussion (e.g., lines 385–412) are overly long and could be streamlined for clarity.
  • Ensure uniform notation of packaging codes (BT, BBFA, VE1, VE2, CC, BBSV, BBCV, BTCV) in all figures, tables, and captions.
  • Verify all references; the list is extensive (lines 446–643) and includes many recent works, but consistency in formatting should be checked.

Recommendation

The manuscript contains valuable experimental data and has the potential to contribute meaningfully to the understanding of polyphenol stability in coffee. However, in its current form, it suffers from methodological weaknesses, incomplete reporting of packaging parameters, and several logical inconsistencies.

I therefore recommend Major Revision. The authors should address the methodological limitations, provide more rigorous statistical treatment, and improve the clarity of the Introduction and Discussion before the manuscript can be reconsidered for publication.

Comments on the Quality of English Language

The manuscript is generally well written, but the English could be improved for clarity and precision. Some sentences are overly long and difficult to follow, while certain terms (e.g., “Mallard reactions”) are misspelled or imprecise. Minor grammatical corrections and stylistic refinements would enhance readability and ensure clearer scientific communication.

Round 2

Reviewer 1 Report

Comments and Suggestions for Authors

I have reviewed the authors' replies to my prior remarks and suggested changes in the manuscript. The authors have adequately addressed all my issues. The edits have enhanced the clarity, accuracy, and discipline of the text, and I commend their carefulness in addressing each point of contention. The manuscript now conforms to the journal's publication requirements, and I endorse its acceptance in its present version.

Reviewer 2 Report

Comments and Suggestions for Authors

The authors have carefully addressed all reviewer comments, and the manuscript has been substantially improved. The revised version presents a clear, well-structured, and scientifically robust study. The introduction now clearly identifies the research gap and contextualises the novelty of combining different coffee ripeness stages and packaging systems. The figures and tables are clear and effectively support the findings.